# The Role of Nucleases Cleaving TLR3, TLR7/8 and TLR9 Ligands, Dicer RNase and miRNA/piRNA Proteins in Functional Adaptation to the Immune Escape and Xenophagy of Prostate Cancer Tissue

**DOI:** 10.3390/ijms24010509

**Published:** 2022-12-28

**Authors:** Gordana Kocic, Jovan Hadzi-Djokic, Miodrag Colic, Andrej Veljkovic, Katarina Tomovic, Stefanos Roumeliotis, Andrija Smelcerovic, Vassilios Liakopoulos

**Affiliations:** 1Department of Biochemistry, Faculty of Medicine, University of Nis, 18000 Nis, Serbia; 2Serbian Academy of Sciences and Arts, 11000 Belgrade, Serbia; 3Department of Pharmacy, Faculty of Medicine, University of Nis, 18000 Nis, Serbia; 4Division of Nephrology and Hypertension, 1st Department of Internal Medicine, AHEPA Hospital, School of Medicine, Aristotle University of Thessaloniki, 54636 Thessaloniki, Greece; 5Department of Chemistry, Faculty of Medicine, University of Nis, 18000 Nis, Serbia

**Keywords:** prostate cancer, Toll-like receptors, acid nucleases, immune escape, xenophagy

## Abstract

The prototypic sensors for the induction of innate and adaptive immune responses are the Toll-like receptors (TLRs). Unusually high expression of TLRs in prostate carcinoma (PC), associated with less differentiated, more aggressive and more propagating forms of PC, changed the previous paradigm about the role of TLRs strictly in immune defense system. Our data reveal an entirely novel role of nucleic acids-sensing Toll-like receptors (NA-TLRs) in functional adaptation of malignant cells for supply and digestion of surrounding metabolic substrates from dead cells as specific mechanism of cancer cells survival, by corresponding ligands accelerated degradation and purine/pyrimidine salvage pathway. The spectrophotometric measurement protocols used for the determination of the activity of RNases and DNase II have been optimized in our laboratory as well as the enzyme-linked immunosorbent method for the determination of NF-κB p65 in prostate tissue samples. The protocols used to determine Dicer RNase, AGO2, TARBP2 and PIWIL4 were based on enzyme-linked immunosorbent assay. The amount of pre-existing acid-soluble oligonucleotides was measured and expressed as coefficient of absorbance. The activities of acid DNase II and RNase T2, and the activities of nucleases cleaving TLR3, TLR7/8 and TLR9 ligands (Poly I:C, poly U and unmethylated CpG), increased several times in PC, compared to the corresponding tumor adjacent and control tissue, exerting very high sensitivity and specificity of above 90%. Consequently higher levels of hypoxanthine and NF-κB p65 were reported in PC, whereas the opposite results were observed for miRNA biogenesis enzyme (Dicer RNase), miRNA processing protein (TARB2), miRNA-induced silencing complex protein (Argonaute-AGO) and PIWI-interacting RNAs silence transposon. Considering the crucial role of *purine and pyrimidine* nucleotides as *energy* carriers, subunits of nucleic acids and nucleotide cofactors, future explorations will be aimed to design novel anti-cancer immune strategies based on a specific acid endolysosomal nuclease inhibition.

## 1. Introduction

Prostate cancer (PC) represents the leading cause of cancer-related deaths in males, with increasing incidence and a low cure rate. It represents the fifth leading cause of cancer-related death in men worldwide [1,2,3]. Due to highly aggressive properties, rapid proliferation, high metastatic potential and resistance to therapy, PC remains a challenge in elucidating the immune escaping mechanisms and the corresponding biomarkers [3,4]. Alterations in miRNA expression profiles in tumors may result in a reduced functional impairment of natural killer cells, macrophages and dendritic cells, all of which are the main components of the innate immune response [5].

The prototypic sensors for the induction of innate and adaptive immune responses are Toll-like receptors (TLRs) [6]. In addition to their typical residence in immune cells, a remarkably high expression of NA-sensing TLRs has been found in PC cells and across a wide range of different tumors [1]. They recognize both the pathogen associated molecular patterns (PAMP) and the damage related molecular patterns (DAMP). Among human nucleic acid-sensing (NA-sensing) TLRs are: the TLR3, which recognizes viral double-stranded RNA and synthetic polyinosinic-polycytidylic acid (poly I:C) fragments; the TLR7/TLR8 which recognize fragments of single-stranded RNAs, miRNAs, synthetic imidazoquinoline derivatives, guanosine- and uridine-rich ssRNA and synthetic polyuridines (poly U) [1,7]; the TLR9, which detects double stranded DNA (dsDNA), unmethylated CpG motifs of DNA from viral and bacterial origin and synthetic unmethylated CpG dinucleotides [8]. High expression of TLR9 in PC cells has been associated with less differentiated, more aggressive and more propagating forms of PC [9]. High expression of TLR7/TLR8 in lung cancer cells has been accompanied by resistance to apoptosis induced by chemotherapy and increased tumor cell survival [10].

The endosomal nucleases compartmentalized in endolysosomes, have been recognized as the upstream enzymes which degrade exogenous and endogenous DNAs and RNAs, hence generating oligonucleotides capable of acting as TLR ligands. In this way, endonucleases, such as RNase T2 and DNase II, may critically contribute to the activation of NA-TLRs and subsequent downstream signaling, associated with cytokine secretion [11,12,13].

However, there are no results about the catalytic activity of endosomal/lysosomal nucleases in PC cells and their role in PC carcinogenesis. Accordingly, the expectations of the fundamental role of endosomal/lysosomal nucleases may be empirically accurate as hypothesis of xenophagy and immune-escaping mechanisms within PC tissue.

The supply and digestion of surrounding metabolic substrates from dead cells represents a specific mechanism of cancer cells survival, defined as cannibalism, xenophagy or xeno-cannibal activity. An unusual, phagocytic-like acquired scavenging activity of cancer cells has been associated to their invasiveness, called the “escamotage“ property. Among the well-recognized substrates for xenophagy are proteins, rapidly degraded by intracellular catepsins [14]. Actually, the xenophagy represents the adaptation of tumor cells, to enable metabolic precursors from dying cells for their own survival and proliferation, based on acquired phagocytic properties. It is stimulated in acidic environment [15]. Besides the environmental dead cells, tumor cells are also able to cannibalize live immune cells, such as CD8+ lymphocytes, contributing to the immune escaping [16,17,18].

The enzymes responsible for acidic hydrolysis of RNAs and different-sized oligonucleotide fragments belong to the nuclease family, hydrolyzing the phosphodiester bonds within the ribonucleotide, at pH about 5 [19]. One of the most precisely defined is the Ribonuclease T2 (RNase T2). The tissue expression of gene cluster of acid RNase (*RNase T2)* has been detected in prostate, while the enzyme has been detected almost in all tissues, which is documented also by the Cancer Genome Atlas database [20]. The enzymes accountable for hydrolysis of DNA at acidic pH optimum (pH 4.8–5.2) are DNase II-α, DNase II-β and L-DNase II, commonly known as DNase II [21]. The DNase II fragmentation of exogenous DNA is a precondition of TLR9 ligand sensing [6]. Besides the fragmentation of DNA from exogenous origin, it also degrades the apoptotic bodies containing DNA. Prostate is very rich in DNase II-β from which it can be secreted in extracellular space [21].

The role of specific RNases in biogenesis and maturation of non-coding microRNAs (miRNAs) and small interfering RNAs (siRNAs), involves a terminal catalytic activity of RNase III, known as Dicer ribonuclease and PIWI proteins. Dicer RNase is involved in terminal cleavage of pre-miRNAs with hairpin-like structures into last miRNA structures, functionally active with 21–23 nucleotides in length [22,23]. Decreased expression of Dicer RNase has been documented in ovarian cancer, correlating with advanced tumor stage. Pleiotropic role in PC makes it interesting and still enigmatic. The destruction of Dicer RNase expression subdues the growth of human prostate cancer cell lines, while it induces the increased migration and invasion of PC cells [23]. The crucial roles of specific proteins in miRNA processing (TARB2 protein) and in miRNA-induced silencing complex (Argonaute proteins, AGOs) have been documented [24,25]. The PIWI proteins, presumably required for spermatogenesis, are capable of interacting with small non-coding RNAs, known as the PIWI-interacting RNAs (piRNAs) silence transposons. PIWI protein also associates with some miRNAs. It exhibits a general preference for uridine-rich sRNAs. Its altered expression, together with the altered formation of piRNAs, has been documented in a number of cancer types [22,26]. Tumor cells perform tumor immune escape by atypically expressing related miRNAs, which significantly impair the anti-tumor immune response by manipulating many components of both innate and adaptive immunity. Furthermore, they inhibit apoptosis of tumor cells by acting on various proteins involved in the apoptotic pathways [5].

The understanding of the importance of endolysosomal degradation of foreign-derived and host-derived nucleic acid, as a prerequisite for corresponding ligands sensing by the NA-TLRs, their processing, utilization and downstream effects on PC tissue, are still elusive. In this study, we set out to explore the activity of acid DNase II and RNase T2, the activities of nucleases toward TLR3, TLR7/8 and TLR9 ligands (Poly I:C, poly U and unmethylated CpG), the NF-κB p65 and acid-soluble nucleotides as potential novel markers of healthy tissue transition into malignant phenotype and PC cell survival. In order to explore the importance of altered biogenesis, maturation and processing of miRNAs in PC, Dicer RNase and the miRNA/piRNA assembled proteins (AGO2/TARBP2/PIWI) were examined as well.

## 2. Results

The age and the level of standard histopathological and biochemical biomarkers of PC are shown in Table 1. The values above 4 ng/mL of PSA were suspicious for cancer.

### 2.1. Enzyme Assays

An approximate fourfold increase of specific enzyme activity *in tumor tissue* was referred for *RNase T2 in RNA degradation (Figure 1**A) and degradation of specific NA-TLR3/7/8 ligands (**Figure 1B,C) in the following order: RNA > Poly I:C > poly U*.

*The activity of DNase II increased almost four times in tumor tissue. It was* followed by about double increase of its activity in adjacent carcinoma tissue, compared to the control healthy tissue counterparts *(Figure 1D). The activity of TLR9 ligand unmethylated CpG dinucleotidase increased almost two times (Figure 1E)*.

The expression pattern of Dicer RNase was suppressed in tumor tissue, compared to control healthy and tumor adjacent tissue, what was shown on Figure 1F.

A noteworthy increase in acid-soluble nucleotide specific absorbance was observed in PC tissue (Figure 2A). The level of hypoxanthine was considerably higher in PC tissue, compared to tumor adjacent and the control healthy tissue. Contrary to hypoxanthine, the concentration of xanthine and uric acid was significantly lower in tumor tissue (Figure 2B).

### 2.2. Proteomic Analysis

We afterward examined the level of NF-κB p65 (Log. Ext). A considerably increased level of NF-κB p65 active subunit was documented in cancer specimen tissue, compared to the corresponding healthy tissue. No difference was observed between tumor and tumor adjacent tissue (Figure 3A).

The level of AGO protein (pg/mL) significantly decreased only in tumor adjacent tissue, but not in tumor tissue (Figure 3B). The level of TARBP2 (pg/mL) significantly decreased in tumor and tumor adjacent tissue, compared to the control tissue (Figure 3C). The level of PIWI protein (pg/mL) significantly decreased in tumor adjacent tissue, compared to the control tissue (Figure 3D). No substantial difference was registered between tumor and tumor-adjacent tissue.

Multiple stepwise regression analyses, including all variables that were correlated with PSA (Gleason score, TARBP2, acid DNase specific activity and healthy tissue acid soluble nucleotides) showed that in univariate analysis, both DNase II and TARBP2 were associated with PSA (*p* = 0.019, B = 3.55, 95%CI = 0.6–6.5 and *p* = 0.042, B = −0.07, 95%CI = −0.14 to −0.003, respectively). However, in multivariate analysis, only DNase II specific activity remained a significant positive predictor of PSA (*p* = 0.013, B = 3.41, 95%CI = 0.78–6.05). Regression coefficients (β) of stepwise multiple regression models with dependent variable PSA and independent variables selected based on respective correlations were determined. Highly positive correlation was observed between the PSA level and acid DNase II. Significant positive correlation was observed between the PSA level and acid soluble nucleotides only in the healthy control tissue. Significant negative correlation was observed between the PSA level and TARBP2 protein.

## 3. ROC (Receiver Operating Characteristic) Curves

According to the Biomarker detection group, the molecular biomarker evaluation implies its biologic sensitivity and specificity to the disease based on reliable and accurate measurement method [27]. Our previous experience encompassed the reference measurement procedures performed in order to define the range of cancer test values. To further investigate the possible predictive value for PC development of investigated parameters, we performed ROC curve analysis (Figure 4). Evaluation of areas under the curves (AUCs) showed that the enzymes RNase T2, DNase II, TLR9 CpG, TLR3 (Poly I:C) and TLR7/8 (poly U) specific enzyme activities predicted PC in a significantly high performance. After determining the optimal cut-off values by Youden’s index, we calculated the sensitivity and specificity and we found that RNase T2 identified PC with a sensitivity of even 100% and a specificity of 92.5%, while the sensitivity and specificity of DNase II were 97.5% and 88.8%, respectively (Table 2).

## 4. Discussion

This is the continuation of our study previously published in which we showed that a significantly upregulated Poly(A) deadenylase activity in PC and tumor-adjacent tissue is a promising RNA-protein bypass biomarker of PC development, while RNASEL may predict chronic prostate inflammation [28].

Performed study documented about a fourfold increased activity of acid RNase T2 and DNase II in cancer tissue, compared to the corresponding control tissue. The activities of nucleases toward TLR3, TLR7/8 and TLR9 ligands (Poly I:C, poly U and unmethylated CpG) have been increased too (about 2–3 times). In order to employ the observed enzymes as possible surrogate markers of healthy tissue transition into malignant phenotype, they have been detected in the adjacent surgically healthy tissue as well. Compared to the corresponding healthy counterparts, distant from carcinoma 2 cm at least, the activity in adjacent surgically healthy tissue was almost two times higher. In line with the results observed, we may hypothesize that acid RNase T2 and DNase II may serve as early markers for malignancy, even in the case that the transition (adjacent healthy tissue) is not so profound to be documented in macroscopic or histopathological examination in tumor adjacent tissue. The activity of DNase II correlated with the traditional marker PSA.

In order to understand the complexity of a paradigm as a way to conceptualize the knowledge about the role of acid nucleases in PC, it is essential to mention the potential substrates. The excessive amount of NAs are released into tissue extracellular space in the form of free NAs, exosomes and microvesicles, as a result of tumor tissue necrosis, necroptosis and apoptosis, in response to starvation, or gene damage. They act as the DAMP signals, where NA-cleavage enzymes create different-sized NA-TLR ligands, which are sensed through interaction with the corresponding TLRs (3, 7/8 and 9) after tumor cells endocytosis [13]. The NA-sensing TLRs reside in the endoplasmic reticulum, endosomes and lysosomes [7]. Following NA binding, TLRs undergo conformational changes allowing the homophilic binding between the TIR domains with specific adapters, TRIF or MyD88. Endosomal TLR3-ligand binding stimulates TRIF adaptor, which associates TRAF3, TRAF6, RIP-1 and IRF-3, leading to final activation of inflammatory transcriptional factor NF-κB, mitogen-associated protein kinase (MAPK) and interferon-β (IFN-β). Endosomal TLR7/8/9 use MyD88 as the first adaptor protein. From one side, the activation of TLR7/9 leads to the stimulation of IRF proteins through the engagement of IRAK1/4, TRAF3, and TRAF6, followed by secretion of type I interferons. The activation of NF-κB, after the dissociation of its inhibitory part IκB and subsequent translocation of NF-κB p65 unit to the nucleus, IRFs and AP-1 transcription factors elicit transcription of proinflammatory cytokines [1,7]. In PC, the activation of TLRs induces the liberation of tumor-promoting cytokines and chemokines, such as IL-1, IL-6, IL-7, IL-8, IL-10, IL-13, IL-17, IL-23, TGFβ, CXCL8, CXCL1, CXCL12, CCL2, CX3CL1, CXCL1, CXCL16 and VEGF. They induce chronic “sterile inflammation”, which contribute to the cancer invasiveness and metastasis via extracellular matrix remodeling [29].

The receptor–ligand complex is accountable for transducing proper information about ligand concentration in the cell, as it was excellently elaborated in mathematical modeling by Shankaran et al. [30]. Our hypothesis is that the rapid endo-lysosomal degradation of NA-TLR ligands by specific, abovementioned acid nucleases in PC tissue can evade acute attenuation of TLRs (endocytic downregulation) and ligand-induced receptor desensitization, thus maintaining constant TLR receptor expression. In fact, the expression pattern of NF-κB p65 has not been linearly related to the available ligands input. Almost equally important is the fact that the endolysosomal degradation of NA enables useful nutritional support for tumor cells [31]. Contrary to that, the failure to clear NA properly by DNase II leads to autoimmunity, which has been recognized in some autoimmune diseases, such as lupus erythematosus [32].

The xeno-cannibal activity seems to be particularly important for purine nucleotides, the central molecules for nucleic acid synthesis, energy production (ATP, GTP), and cell signaling (cAMP, cGMP), since the de novo synthesis of purine nucleotides is a “privilege” of a small number of tissues in the body, due to its complexity and a considerable number of precursors needed. In such conditions, the salvage purine reutilization pathway from preexisting purine bases seems to be the only alternative to tumor cell survival. Hence, our hypothesis about the role of endosomal TLRs in the reutilization of purine and pyrimidine nucleotides may be justified.

In this way, PC may “kill two birds with one stone”—acquiring their own immune escaping system and supporting purine/pyrimidine available intracellular pool. The presence of free oligonucleotides, nucleosides and purine bases should maintain metabolic and cellular homeostasis for energy demands (ATP and GTP) and nucleic acids synthesis through the purine salvage pathway, presumably via hypoxanthine reutilization [33]. Our results (Figure 2) proved significantly higher level of hypoxanthine in PC tissue, compared to tumor-adjacent and the control healthy tissue.

Among the immune defense cells surrounding tumor tissue are tumor-associated macrophages (TAMs), tumor-associated neutrophils (TANs) and mesenchymal stem cells (MSC). During tumor progression, TAMs change their functional properties, from anti-tumor M1-polarized state, able to kill cancer cells, to an M2, pro-tumor state, which tolerate the tumor cells, by secreting tumor-promoting cytokines and by recruitment of immunosuppressive MSC [34]. Inappropriate host immune response may develop the anti-tumor immune tolerance. Having in mind the central role of NA-sensing TLRs in immune defense cells, the intervention via exogenous NA-TLRs ligands has been employed in vaccination-based strategies of cancer immunotherapy [8,35,36]. For that purpose, the NA-sensing TLRs target ligands have been designed to stimulate anti-cancer immune cell response. Among them are the CpG-siRNA conjugates, Poly I:C and polyU nuclease-resistant therapeutic molecules. In order to escape acid nuclease activity and to prolong the retention time in immune cells, they have been modified (CpG phosphoro-thioate linkages, inorganic hybrid nanostructures, Poly I:C stabilized with poly-L-lysine and carboxymethyl-cellulose, imidazoquinoline derivative resiquimod, TMX-101) [35,37]. The preclinical and clinical trials with the introduction of NA-TLR targeted ligands have been performed with variable success [38,39,40].

Taken together, the data reported in our study emphasize the particular importance of nucleic acids and of NA-TLR3/7/8/9 ligands degradation in PC. The potential aim of improving therapeutic protocols of PC treatment may be occurred via modulating the activities of mentioned acid nucleases. A large number of various compounds, different in origin (natural or synthetic), may act as DNase inhibitors, some specific toward only one type or active toward more types of DNase. An overview of natural and synthetic DNase II inhibitors is given in our previous review [41]. Detection and design of novel DNase II as well as RNase T2 inhibitors might be the field of novel objectives and efforts. Downregulation of pH is a known cancer mark. Acidity drives cancer invasion, toxic for normal but not for cancer cells [42]. Our results are in line with the data supporting that acidic tumor environment can favor progression and drug resistance of cancer [43]. Endolysosomal TLR3/7/8/9 responses are diminished by substances that induce alkalinization, such as macrolide antibiotic bafilomycin A1, chloroquine, or ammonium chloride [44]. By targeting lysosomes, they can inhibit apoptosis and cell recycling, responsible for the clearance of organelles, proteins and NAs. Having in mind that inflammatory agents may induce PC, and that prostate inflammation may be one of the contributing risk factors, potential importance of antimicrobial agents capable of affecting authophagy should not be ignored.

Some experimental results and recent clinical study have supported our hypothesis of the crucial role of acidic nucleases, where alkalization of the tumor microenvironment, through bicarbonate administration, has been associated with improved results of cancer treatment, improved survival and suppressed metastasis development [43].

An increase of NF-κB p65 unit expression in cancer tissue, more than in tumor adjacent tissue has been reported (Figure 3A). However, its upregulation has not been proportional to NA-sensing TLRs ligand specific nucleases activity. Elevated NF-κB may predict to worse prognosis, progression and therapy resistance in PC patients, acting as a direct downstream target of the Akt signaling cascade [44]. In support of our results are experimental in vitro studies, which reported that NF-κB activates acid phosphatase (PacP) promoter activity gene expression in PC [45]. With regard to the cytosolic (self) nucleic acids, specific sensors are RIG-I-like receptors (RLRs), followed by the stimulation of cGAS/STING, DAI, IFI16 and DDX41 pathway. Despite different primary compartment for NA sensing, common downstream cascade is completed by the activation of NF-κB and transcription of proinflammatory cytokines [13,46].

MiRNAs represent an *ubiquitous* cellular machinery in human cells, including tumor cells, to regulate cell proliferation, apoptosis and control of immune responses. They are transported among the extracellular space in extracellular microvesicles and exosomes and interact with TLR7/8 [47]. Similar effect has been described for uridine-rich (polyU) single-stranded RNA synthetic analogues, which represent powerful ligands for human TLR7/8 [48]. Approximately three times increased nuclease activity toward polyU (Figure 1) may suggest that TLR8 may be actively involved in the extracellular miRNA-sensing and subsequent degradation in PC. In assessing the functional profile of Dicer RNase and its related miRNA (AGO2 and TARB4) and piRNA (PIWI) assembled regulatory proteins of PC, our study documented the fall of Dicer RNase in PC specimens, followed by gradual loss in tissues adjacent to cancer, compared to the corresponding control healthy tissue. It was connected with a significant fall in the TARB2 level. The level of AGO2 and PIWI declined in tumor adjacent tissue (Figure 3). Obtained results may suggest an arranged reprogramming of cancer tissue miRNA-related enzymes of their synthesis and assembled regulatory proteins. The pleiotropic roles of Dicer RNase in cancer were described as oncogenic, tumor suppressor and metastatic regulator. Considering the Dicer RNase as the check-point in miRNA synthesis from their pre-miRNAs precursors, the results about reduced or missed Dicer RNase expression in breast and lung cancer tissue are in accordance with our results [23,49]. Our results are in accordance with the impaired miRNA biogenesis, maturation and processing in cancer tissues [49,50]. As a result, 42 protein framework complexes have been dysregulated in PC progression, involved in post-transcriptional gene silencing of targeted mRNAs, via specific binding to complementary nucleotides of the target mRNA [45]. There have been designed the master oncomirs in PC: miR-1, miR-16, miR-21, miR-106b, miR125b, miR-141, miR-145, miR-155, miR-221, and miR-375 [2,51]. PIWI proteins exert germline-restricted functions, involved specially in spermatogenesis and regulation of germline stem cells. The depletion of DICER1 may lead to decreased PIWI protein association and fall in spermatogenesis [22].

The limitation of the study is an insufficient sample size for proposing acid nucleases as the markers for transition of normal prostate tissue to a malignant one.

## 5. Materials and Methods

The Ethical review board of University Clinical Center Medical Faculty Nis approved this prospective study protocol and waved informed consent (No.12-8818-2/18).

### 5.1. Patient Selection

Our pilot study was conducted at University Clinical Center Nis, where 40 consecutive patients with clinically verified localized PC underwent surgical radical prostatectomy. These were primary lesions with no neoadjuvant treatment. The diagnosis and staging of PC has been verified by clinical symptoms, abnormal findings on digital rectal examination (DRE), increased age-specific reference range of PSA, TNM staging, based on T (primary tumour size), N (spread of cancer to nearby lymph nodes) and M (metastases of cancer to distant sites) and by Gleason score grading (6 is a low grade, 7 is an intermediate grade, and 8–10 as a high grade) [52]. PSA as a biomarker is organ specific, but not cancer specific, with normal reference range of less than 4 ng mL^−1^, with a diagnostic grey area between 4 and 10 ng/mL [53]. All tissue patterns are categorized by Gleason grading system by using histopathological grading scheme for prostatic adenocarcinoma.

### 5.2. Tissue Preparation

After surgical prostatectomy, a part of the cancer tissue, adjacent surgically healthy tissue and normal tissue counterpart, at least 2 cm far from the carcinoma site, were dissected. No appearance of microfocal carcinomas in surgically healthy tissue at least 2 cm far from the carcinoma site was documented, confirmed by the histopathological examination of all tissue slices. Obtained samples were homogenized on ice, 10% homogenates were prepared and stored at −80 °C until biochemical analysis.

### 5.3. Enzyme Assays

The protocols used for the determination of the activity of RNases and DNase II have been optimized in our laboratory, as described before for tissue and cell culture samples, and for plasma specimens [54,55].

It is based on spectrophotometric measurement of released acid soluble nucleotides at 260 nm, from the corresponding enzyme substrates, (RNA, Poly I:C, Poly(U), DNA and CpG) in acidic conditions of pH = 5. The subtraction of pre-existing acid-soluble oligonucleotides was performed by simultaneous analysis of the corresponding control samples, where the corresponding substrates were added after cold precipitation and centrifugation at 6000× *g* for 15 min at +4 °C. Enzyme activities have been expressed as total activity (U/L homogenate) according to the proposed calculation of enzyme Units (1U is equal to 0.1 ABS), by Akagi et al. [56].

The specific activity has been expressed as IU/g protein. Tissue protein content in homogenates was measured according to the Lowry procedure, based on the reaction of tissue proteins with an alkaline copper tartrate and Folin reagent [57].

The protocol used to determine Dicer RNase (ribonuclease III) was based on enzyme-linked immunosorbent assay (kits were purchased from MyBio source San Diego, CA, USA) with the detection range from 0.312 to 20 ng/mL.

The protocols used to determine human AGO2, human TARBP2 (RISC-loading complex subunit TARBP2) and human PIWIL4 (Piwi-like protein 4) were based on enzyme-linked immunosorbent assay (kits were purchased from Wuhan Fine Biotech Co., Ltd. Hubei, China) with the detection range for AGO2 from 15.625 to 1000 pg/mL; for TARBP2 from 0.313 to 20 ng/mL; for PIWIL4 from 15.625 to 1000 pg/mL.

The amount of pre-existing acid-soluble oligonucleotides was measured in each sample and expressed as coefficient of absorbance (Abs). The levels of purine bases (hypoxanthine, xanthine) and uric acid were measured by the method of Dudzinska and Lubkowska [58] and expressed in their percentage part of a whole concentration.

### 5.4. Determination of NF-κB p65 in Prostate Tissue Samples

An enzyme-linked immunosorbent method was standardized in our laboratory according to the previous protocol published for liver tissue [59].

Prostate homogenate samples were immobilized on solid polystyrene microtiter plates by pipetting 20 μL of each tissue homogenate (equivalent to 2 mg of fresh tissue) and 90 mL of 100 mM carbonate buffer (pH 9.6) into flat-bottomed, 96-well plates, using 2 wells per each sample. The samples were incubated at 4 °C overnight. In *antigen-binding* phase, each test plate was incubated with primary antibody for NF-κB p65 (C-20 sc-372 epitope mapping at the C-terminus of NF-κB, mouse monoclonal IgG1 purchased from Santa Cruz biotecnology) for the next 24 h at 4 °C. Blocking solution of 2.5% BSA in PBS was added afterwards. The plates were washed 3 times with PBS and then incubated with the corresponding fluorescein isothiocyanate (FITC)-conjugated anti-mouse secondary antibody (sc-7972 FITC) for the next 2 h in the dark. The excess of antibody following staining was washed again three times with PBS. Fluorescence measurement was performed on SpectraMax^®^ M Series Multi-Mode Microplate Reader. The mean fluorescence intensity (logarithmic scale) was determined by using the BSA-coated wells to verify the primary and secondary antibody specificity.

### 5.5. Statistical Analyses

All data were tested for normality with Kolmogorov–Smirnov test. We expressed non-normally distributed continuous variables as median with interquartile range, normally distributed continuous variables as mean ± S.D. To identify the differences of continuous variables between the three tissues (tumor, tumor adjacent and healthy tissue), we used independent *t*-test for normally and Mann–Whitney test for non-normally distributed variables, accordingly. Likewise, the differences of continuous variables across the TNM categories were assessed, accordingly. Bivariate associations between continuous variables were examined using Spearman’s rho correlation co-efficient. Correlation analysis of PSA with the following variables: AGO, PIWI, TARBP2, Dicer RNase, NF-κB p65, specific nuclease activity toward ligands for TLR3 (Poly I:C), TLR7/8 (poly U), TLR9 (CpG), RNase T2, DNase II and acid-soluble nucleotides was conducted on an exploratory basis.

Stepwise multiple regression analysis with PSA as the dependent variable and independent variables the variables that were correlated with PSA in bivariate analysis, was used to determine possible predictors of PSA. To further examine the possible predictive ability of the specific enzyme activities, the predictive ability of proteins, the predictive ability of NF-κB p65 and acid-soluble nucleotides that were measured for PSA, we performed receiver operation curves (ROC). In conjunction with the ROC curves, we calculated the Youden’s index, to determine the optimal cut off value of the proteins that predicted PSA. Then, we calculated sensitivity and specificity of these enzymes, proteins and acid-soluble nucleotides to predict prostate cancer. All statistical analyses were performed by the standard IBM statistical Package for Social Sciences-SPSS 18.0 for Windows, Chicago, IL, USA.

## 6. Conclusions

Our data suggest the emerging role of endolysosomal nucleic acids and TLRs-ligands targeted nucleases, serving as prerequisites for PC survival and possible novel cellular biomarkers for PC progression. Obtained results may be employed to design novel anti-cancer immune strategies based on the inhibition of acid TLRs-ligands targeted nucleases.

## Figures and Tables

**Figure 1 ijms-24-00509-f001:**
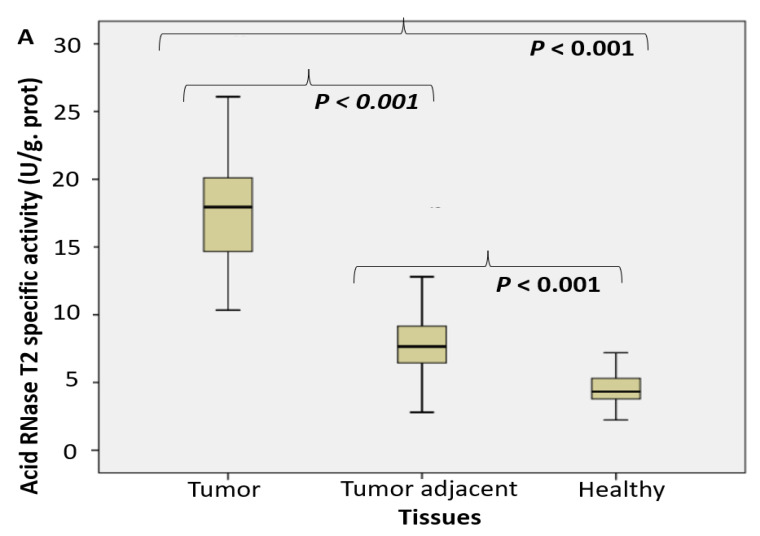
The endosomal acid nucleases activity toward the RNA, DNA, NA-sensing TLRs ligands and Dicer RNase expression profile in PC. (RNase T2 (**A**), Toll3 Poly IC (**B**), Toll 7/8 poly U (**C**), DNase II (**D**), Toll 9 CpG (**E**) specific enzyme activity (U/g prot) and Dicer RNase (pg/mL) (**F**) in prostate cancer, tumor adjacent and healthy tissue, accordingly).

**Figure 2 ijms-24-00509-f002:**
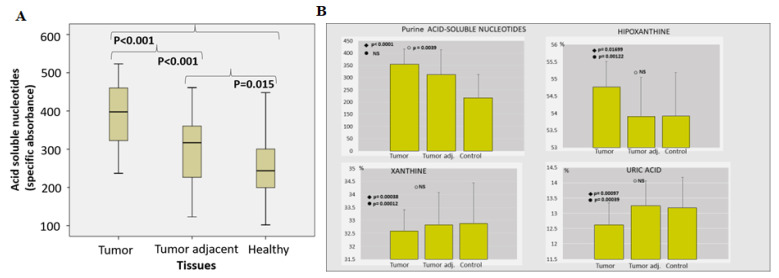
The pool of acid-soluble nucleotides in PC. (Acid soluble nucleotides (specific absorbance) (**A**) and percentual share (%) of hypoxanthine, xanthine and uric acid among purine acid soluble nucleotides (**B**) in prostate cancer, tumor adjacent and healthy tissue, accordingly).

**Figure 3 ijms-24-00509-f003:**
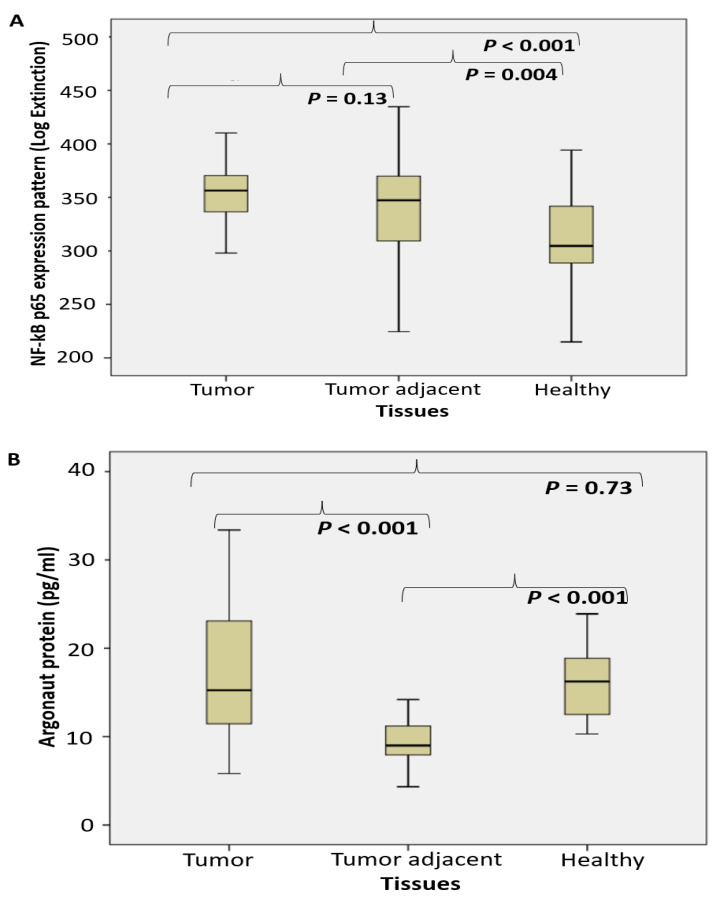
The level of NF-κBp65 and the miRNA/piRNA assembled proteins shift in tumor adjacent tissue of PC. (NF-κB p65 (**A**), argonaut protein AGO2 (**B**), TARBP2 (**C**) and PIWI protein (pg/mL) (**D**) in prostate cancer, tumor adjacent and healthy tissue, accordingly).

**Figure 4 ijms-24-00509-f004:**
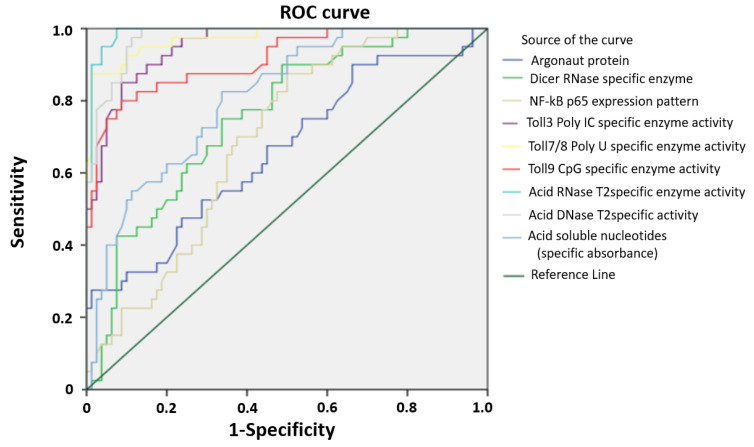
Receiver operating characteristic curve (ROC) of studied enzymes and proteins in PC. ROC showing the performance of studied enzymes and proteins, in patients with PC.

**Table 1 ijms-24-00509-t001:** The age and the level of PSA, Gleason score and tumor stage in investigated patients with PC.

Investigated Parameters	X ± SD
Age	67.05 ± 4.35 (range 26–73 years)
tPSA (ng/mL)	17.26 ± 9.88 (range 5.42–36.45)
Gleason score	6.69 ± 0.55 (range 6–8)
Tumor stage (TNM)	2.33 ± 0.48 (range 2–3)

**Table 2 ijms-24-00509-t002:** Areas under the curves, the optimal cut-off values by Youden’s index, sensitivity and specificity of studied enzymes and proteins in PC (values above 90% are marked red).

Marker	AUC	95% CI	*p*	Cut-Off Value	Sensitivity (%)	Specificity (%)
RNase T2 specific enzyme activity	0.99	0.98–1.00	<0.0001	10.3	100	92.5
DNase II specific enzyme activity	0.98	0.96–1.00	<0.0001	1.65	97.5	88.8
TLR9 (CpG) specific enzyme activity	0.92	0.86–0.97	<0.0001	2.49	80	91.3
TLR3 (Poly I:C) specific enzyme activity	0.95	0.92–0.99	<0.0001	12.8	85	91.3
TLR7/8 (Poly U) specific enzyme activity	0.97	0.95–1.00	<0.0001	0.98	97.5	68.8
Argonaut AGO2 protein	0.66	0.56–0.78	0.004	22.8	27.5	96.3
NF-κB p65 expression pattern	0.69	0.60–0.79	0.001	326	87.5	48.8
Dicer RNase specific enzyme	0.76	0.67–0.84	<0.0001	695	90	51.3
Acid soluble nucleotides (specific absorbance)	0.81	0.73–0.89	<0.0001	314	82.5	66.3

## Data Availability

The data used to support the findings of this study are available from the corresponding author upon request.

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
