# Peer review of "The Role of Nucleases Cleaving TLR3, TLR7/8 and TLR9 Ligands, Dicer RNase and miRNA/piRNA Proteins in Functional Adaptation to the Immune Escape and Xenophagy of Prostate Cancer Tissue"

_ijms, 2022, doi:10.3390/ijms24010509_

Round 1
Reviewer 1 Report
The authors presented well-designed paper on TLRs ligands to be employed in future vaccination-based strategies of cancer immunotherapy. Some doubts were listed below. Although it is basic science paper, the cohort is relatively low (n=40 cases) so I would suggest to use term preliminary data or pilot study and be rather cautious when it comes to conclusions.
In my opinion, introduction is far too specific, and should be either partly transferred to discussion or shortened.
The authors should rewrite the subparagraph concerning the aim of the paper to make it clearer rather than summarize the set of experiments.
As for the tissue preparation – was there any pathological control that these (adjacent surgically healthy tissue and normal 141 tissue counterpart, at least 2 cm far from the carcinoma site) were truly healthy tissues? Please see Lines 141-3 and add your comment.
It is worth adding the phrase that these were primary lesions with no neoadjuvant treatment.
Table 1: I would not suggest presenting basic characteristics as the values with SD, rather range. Please refer to your own table 1 in Cancers (10.3390/ cancers14092239).
Table 1: Please comment the high PSA and relatively low Gleason and low T stage (expect to be either high Gleason and/or high T stage at this PSA). Please comment if Gleason result is post surgical specimen or post biopsy? On which grounds were T tumor stage established (post surgical specimen?)
Author Response
Dear Reviewer,
First of all we would like to thank you for your very useful suggestions, which contribute in a great amount to the quality of the article, its better understanding and the emphasizing a main topic and message.
We have followed all your remarks and suggestions and they can be visible in revised article with track changes (uploaded).

Reviewer 2 Report
The authors performed a study involving the prostate cancer pathway. The study gives some new insights. However, there are some comments:
Material and methods:
-describe Lowry's procedure in the text and cite accordingly.
-Elaborate and give a proper citation for the general audience ( why this is considered borderline) ,, The values above 4 ng/ml of PSA were suspicious for cancer,,
Discussions:
-It is known that specific pathogens also induce deleterious effects such as chronic inflammation and overexpression of oncogenes in the host system. Over time, this can increase the host's susceptibility to tumorigenesis. Hence targeting tumors through anti-microbial mechanisms like xenophagy could be a novel strategy for combinatorial anti-cancer therapy. Please elaborate in a few lines and give proper citations.
-regarding biomarkers, there is an NIH group that gives proper criteria for biomarkers; please classify it accordingly; as a suggestion, see https://pubmed.ncbi.nlm.nih.gov/?term=+FDA-NIH+Biomarker+Working+Group and also,, Ankeet Shah, Dominic C. Grimberg, Brant A. Inman. Classification of Molecular Biomarkers. Duke Cancer Institute, Division of Urology, Duke University Medical Center, Durham, United States. SIUJ Volume 1, Number 1 October 2020,,
Conclusions
-as a primary conclusion, the TLRs-ligands targeted nucleases is proposed as a biomarker. However, in the discussion section, the biomarker perspective of TLRs-ligands is not sufficiently discussed -please elaborate in the discussion part.
Author Response

(The authors gave the same response as above.)

Reviewer 3 Report
This paper introduced ‘the role of nucleases cleaving TLR3, TLR7/8 and TLR9 ligands, Dicer RNase and miRNA/piRNA proteins in functional adaptation to the immune escape and xenophagy of prostate cancer tissue’. It is a topic of interest to the researchers in the related areas, but the paper needs significant improvement before acceptance for publication. My detailed comments are as follows:
1. Grammar: Only some proofreading has been done on the introduction part (For example, mistakes in lines 31,48,50,56,61…), and many grammar and spelling problems have been found. It is suggested that the author proofread the whole grammar and words, and if necessary, find relevant institutions to polish the language in this article.
2. Abstract: It is recommended that all objectives, methods, results, and conclusions be explicitly included, with detailed data supporting the results section.
3. Introduction and Conclusion: The author provided detailed background information and introduced the author's research content. But does this research have potential research value? For example, what is the specific incidence of prostate cancer, and the specific proportion of people who get sick and die from this disease every year?
4. Methods and conclusions: The introduction of research methods is clear, the structure and layout of the article are simple and easy to understand, and it can prove that the views put forward by the author are correct and effective. The research is supported by appropriate theories and literature. The method, experiment, and conclusion in this paper are correct, comprehensive, and reasonable. However, some illustrations in this paper, such as Figure 1 and Figure 2B, are a bit blurry. Please consider proofreading the clarity of the illustrations in the subsequent revision.
5. Discussion: The author has made a complete explanation and analysis of the results, and the conclusion is insightful. The future research direction of the team has been clarified, and this research prospect undoubtedly excites readers. However, does the author neglect the limitations of their research? Every experiment and method has its limitations, and the author is requested to point this out in the subsequent revision.
6. References: The research is supported by appropriate theories and literature. But references before 2010 accounted for one-third of the total references, and it was recommended to reduce the references in this part, and more references after 2010.
Author Response
First of all we would like to thank you for your very useful suggestions, which contribute in a great amount to the quality of the article, its better understanding and the emphasizing a main topic and message.
We have followed all your remarks and suggestions and they can be visible in revised article with track changes (uploaded).

Round 2
Reviewer 2 Report
The manuscript can be published in its present form.
Reviewer 3 Report
accept.